# Toward Lightweight Diabetic Retinopathy Classification: A Knowledge Distillation Approach for Resource-Constrained Settings

Niful Islam , Md. Mehedi Hasan Jony, Emam Hasan, Sunny Sutradhar, Atikur Rahman and Md. Motaharul Islam *

Department of Computer Science and Engineering, United International University, Dhaka 1212, Bangladesh; ehasan201302@bscse.uiu.ac.bd (E.H.); ssutradhar201084@bscse.uiu.ac.bd (S.S.)
* Correspondence: motaharul@cse.uiu.ac.bd

**Abstract:** Diabetic retinopathy (DR), a consequence of diabetes, is one of the prominent contributors to blindness. Effective intervention necessitates accurate classification of DR; this is a need that computer vision-based technologies address. However, using large-scale deep learning models for DR classification presents difficulties, especially when integrating them into devices with limited resources, particularly in places with poor technological infrastructure. In order to address this, our research presents a knowledge distillation-based approach, where we train a fusion model, composed of ResNet152V2 and Swin Transformer, as the teacher model. The knowledge learned from the heavy teacher model is transferred to the lightweight student model of 102 megabytes, which consists of Xception with a customized convolutional block attention module (CBAM). The system also integrates a four-stage image enhancement technique to improve the image quality. We compared the model against eight state-of-the-art classifiers on five evaluation metrics; the experiments show superior performance of the model over other methods on two datasets (APTOS and IDRiD). The model performed exceptionally well on the APTOS dataset, achieving 100% accuracy in binary classification and 99.04% accuracy in multi-class classification. On the IDRiD dataset, the results were 98.05% for binary classification accuracy and 94.17% for multi-class accuracy. The proposed approach shows promise for practical applications, enabling accessible DR assessment even in technologically underdeveloped environments.

**Keywords:** attention; CNN; diabetic retinopathy; knowledge distillation; transformer

## 1. Introduction

The human eye is an exceptional organ that provides us with the priceless benefit of vision, enabling us to connect with and experience the outside world and convey our emotions. A diabetes-related eye disease, termed diabetic retinopathy(DR), however, can impair this crucial skill. Diabetes-related high blood sugar may harm the blood vessels in the retina, impairing eyesight [1]. Currently, this is the most common cause of eyesight loss [2]. Early stages might not present any symptoms, highlighting the value of routine eye exams for early identification. Vision loss or impaired vision may present as symptoms as it advances. Diabetic retinopathy progresses in five phases. The earliest stage of no diabetic retinopathy (no DR) reveals no retinal damage. Microaneurysms occur in mild non-proliferative diabetic retinopathy (mild NPDR), causing minimal visual distortion [3]. Moderate non-proliferative diabetic retinopathy (moderate NPDR) causes vascular obstructions and decreased retinal oxygen, which can lead to the formation of new blood vessels [4]. Severe non-proliferative diabetic retinopathy (severe NPDR) is characterized by substantial oxygen deficiency and increased oxidative stress. Controlling blood sugar, hypertension, and cholesterol through dietary modifications, medication, and insulin therapy are key components of managing and preventing diabetes [5]. To monitor retinal health

and enable prompt actions, routine eye exams are essential [2]. Treatments, including laser photocoagulation, intraocular injections, or vitrectomy surgery, can be necessary in extreme cases to stop future vision loss [6]. It is crucial for individuals with diabetes to take charge of managing their illness and make their sight priority number one in order to decrease the probability and effects of diabetic retinopathy [7]. The use of computer vision technologies in the detection and treatment of diabetic retinopathy offers various benefits. It facilitates early detection and increases screening accessibility [8,9]. It makes unbiased assessments possible for individualized and affordable therapies [10]. Additionally, the data produced support research and facilitate more effective therapeutic approaches.

Convolutional neural networks (CNNs), a sub-field of artificial neural networks, are widely used in computer vision tasks [11]. They are capable of learning hierarchical representations from input images, making them effective for image classification, object detection, and segmentation [12]. CNNs excel at capturing local patterns and spatial dependencies through convolutional layers, while pooling layers reduce spatial dimensions and preserve important information [13]. Despite their high performance, CNNs have limitations when dealing with small datasets. They require significant amounts of labeled data for training, which are often impractical or expensive to obtain. Additionally, training CNN models is computationally demanding and requires powerful hardware and time resources. To overcome these limitations, transfer learning has emerged as a powerful solution. By leveraging pre-trained CNN models, which have been trained on large datasets for generic image recognition tasks, transfer learning reduces the need for massive labeled datasets and lowers the computational burden [14]. The idea is to use the knowledge learned via a pre-trained CNN from one task and apply it to related tasks [15]. Since it starts with a pre-trained CNN backbone and only requires fine-tuning a small portion of the model, the number of training iterations required is significantly reduced, speeding up the training process. Another advantage of transfer learning is its efficiency with a small amount of data. It is particularly useful when faced with limited data, as it leverages knowledge from large-scale datasets to improve performance on smaller datasets [16].

Another significant limitation of CNNs is their inability to effectively capture global context due to their local receptive fields [17]. This is particularly noticeable when dealing with images that contain substantial objects or intricate relationships between remote regions. The Vision Transformer (ViT) [18] addresses this problem by leveraging the transformer architecture's attention mechanism, enabling efficient global information capture. ViT divides an image into fixed-size, non-overlapping patches. These patches are linearly projected by multiplying them with a learnable weight matrix and their positions are embedded with them. Finally, they are treated as a sequence. It utilizes the self-attention mechanism to identify long-distance dependencies, allowing for interactions between patches. ViT is now highly competitive in a variety of image classification tasks [19,20]. ViT, however, still has trouble managing high-resolution images effectively. When applied directly to large images, the self-attention mechanism's quadratic complexity, with respect to the input sequence length, becomes a bottleneck and hinders scalability. The Swin Transformer [21] is useful in this situation. The Swin Transformer uses a hierarchical structure of self-attention mechanisms to handle high-resolution images. The image is split into non-overlapping windows and is processed hierarchically across several layers. As attention is calculated within local windows rather than across the entire sequence, this lessens the quadratic complexity issue. Swin Transformer outperforms some state-of-the-art architectures in various image recognition tasks. Therefore, it is currently dominating the field of computer vision.

The rising complexity and resource requirements of contemporary deep learning models give rise to the need for knowledge distillation [22,23]. These models are difficult to deploy on resource-restricted devices and in situations with constrained processing power as their sizes and computational requirements increase. By transferring the information and insights of a larger teacher model to a smaller student model through knowledge distillation, this problem is resolved [24]. This allows the student model to attain substantial

performance while being more computationally efficient. This knowledge compression improves model interpretability, energy efficiency, and robustness while providing a realistic way to regularize model training. It also enables deployment on edge devices, mobile platforms, and low-resource contexts [25].

In order to address the need for accurate diabetic retinopathy classification using a lightweight image classifier, this article presents a knowledge distillation-based approach. The proposed solution involves a series of image preprocessing tasks to enhance the image quality and amplify the visibility of the interested areas of the retinal image. Subsequently, a robust teacher model is constructed by fusing Swin Transformer and ResNet152V2, which is leveraged to train the student model. The use of Swin Transformer and ResNet152V2 in the teacher model for knowledge distillation is supported by an amalgamation of architectural efficacy and task-specific considerations. Swin Transformer, a relatively recent approach in computer vision, has remarkable abilities; it extracts both the global and local context from images. It is highly suited for tasks like image categorization due to its hierarchical structure and self-attention mechanism. ResNet152V2, a deep form of the ResNet architecture, on the other hand, is known for its dependability and toughness in a variety of computer vision tasks. Combining these diverse architectures improves the process of knowledge transfer by offering the student model access to a wider range of knowledge. The student model is composed of Xception and a modified convolutional block attention module (CBAM). An empirical study of the portable model on the APTOS and IDRiD datasets attests to its effectiveness. In summary, this research paper has the following major contributions.

- The research involves various image preprocessing techniques to enhance image quality and highlight key areas of interest in the image.
- A robust fusion model is proposed that employs Swin Transformer and ResNet152V2; it serves as an instructive guide for knowledge distillation.
- For efficiency, this research presents a compact student model by merging Xception and a tailored CBAM block, which is 102 megabytes.
- The effectiveness of the proposed model is demonstrated by an empirical evaluation on the challenging APTOS and IDRiD datasets, which yielded excellent results, with 100% accuracy in binary classification and a remarkable 99.04% in multi-class classification on the APTOS dataset. Moreover, the accuracies were 98.05% for binary classification and 94.17% for multi-class accuracy on the IDRiD dataset.

The rest of the article is structured as follows. Section 2 presents the recent studies for DR classification, highlighting their contributions and limitations. Section 3 describes the proposed methods, followed by the results obtained in Section 4. Finally, the article concludes in Section 5.

## 2. Related Work

In recent history, several methods have been proposed for DR detection. Usman et al. [2] presented a deep learning and dimensionality reduction-based method for DR classification. The paper experiments with three pre-trained CNN feature extractors—ResNet50, ResNet152, and SqueezeNet1—to extract useful features from retinal images. The extracted features might contain some irrelevant or redundant ones, which are refined via principal component analysis (PCA) and passed into a machine learning classifier for final classification. Alahmadi [26] proposed a feature extraction module that deteriorates the original feature space of a CNN into content and style representation spaces. Spatial attention and texture attention are integrated into the content and style features, respectively. Later, both streams are fused before the final classification. The proposed module achieves 98% accuracy on binary classification and 85% accuracy on multi-class classification on the APTOS dataset. Farag et al. [27] leveraged dense blocks and transition blocks to construct a custom feature extractor for automatic DR detection. The dense block consists of densely connected convolutional layers, where features extracted from previous blocks are propagated to the subsequent blocks. The transition block, on the other hand, consists of point-wise convolution and $2 \times 2$ max-pooling. This layer compresses

the output of the dense block by reducing the number of features and dimensionality. Finally, the extracted features are passed through a convolutional block attention module (CBAM) [28], where the features are refined with channel attention and spatial attention. This paper also experiments with the APTOS dataset and attains 97% and 82% accuracy on binary class and multi-class classification, respectively. Mondal et al. [29] balanced APTOS19 and DIARETDB1 datasets using generative adversarial network (GAN). For classification, they present an ensemble learning method consisting of two state-of-the-art CNN architectures: DenseNet101 and ResNext. Menaouer et al. [30] presented a fusion model made of two visual geometry group (VGG) networks and a custom deep convolutional network to achieve 90% accuracy on the APTOS dataset. Nonetheless, due to the integration of VGG networks, which are recognized for their high parameter numbers, the proposed system becomes infeasible in real-world applications for modest computational platforms. Mungloo et al. [31] performed an investigation on three high-performing image classifiers, termed VGG16, ResNet50, and DenseNet169. To conduct the experiment, they exploited a public dataset (APTOS) and a private dataset (Mauritian) that the authors collected from local hospitals. Given that APTOS is an imbalanced dataset, the authors performed two data augmentation methods—flipping and brightness adjustment—on the training set to balance the dataset, which resulted in an improvement in the classification performance. Among the three models, ResNet50 produced the best classification accuracy. On the contrary, in separate research [32], Das et al. found that ResNet50 is the most overfitted model among the 26 state-of-the-art deep learning models. According to the experiments, EfficientNetB4 is the most optimal model for DR classification, followed by Inception-ResNet-v2 and NASNetLarge in the second and third positions, respectively. Attallah [33] employed four pre-trained CNN models to extract spatial features. In the next stage, redundant features were deleted and passed to the feature integration phase. Finally, the system classified fundus images with the help of three state-of-the-art machine learning classifiers. Although the model achieves a noteworthy performance in retinopathy of prematurity (ROP) classification, the integration of four classifiers in the feature extraction phase makes the system heavy. Mustafa et al. [34] extracted features from DR images using two pre-trained CNN feature extractors: ResNet50 and DenseNet121. The features were then reduced using principal component analysis (PCA) and fed to an ensemble machine learning classifier, composed of AdaBoost and Random Forest for final classification. Raiaan et al. [35] combined three commonly used DR datasets (Messidor-2, IDRiD, and APTOS) to create a merged dataset. With preprocessing, they presented a lightweight CNN architecture for the classification. Although the model outperformed some of the existing architectures, it consumed over 35 million trainable parameters. Attallah [36] presented a method that uses spectral–temporal, spatial, and textural data to detect retinopathy of prematurity (ROP) in fundus images. To capture texture features, it first performs a Gabor wavelet (GW) conversion, after which the data are ready for three already-trained convolutional neural networks (CNNs). The approach uses discrete wavelet transform (DWT) to combine the data after fine-tuning and feature extraction from these CNNs, and then uses machine learning classifiers, like SVM, LDA, and ESD for ROP diagnosis, to guarantee reliable findings. Image preprocessing plays a crucial role in determining the performance of a deep learning algorithm, especially in image classification [37]. The majority of the systems, however, neglect this crucial step. Nonetheless, Ozbay [38] leveraged an artificial bee colony algorithm to segment the lesion regions of retinal images. The segmented mask is then employed to enhance the image quality, specifically focusing on the fundus region. Subsequently, a seven-layer CNN architecture is developed to classify the DR images. This method achieves a remarkable accuracy of 99.66% in classifying five stages of DR.

Many researchers have segmented some regions of the retinal images and only leveraged the segmented image for classification. Saranya et al. [39] employed U-Net to segment the red lesions in the image. The segmented binary image was then passed to a custom CNN architecture for the final classification. Bilal et al. [40] proposed a two-stage DR detection system that first segmented the blood vessels and optic disk employing U-Net.

In the second stage, they integrated Inception-V3 for classification. Jena et al. [8] used U-Net to segment and delete the blood vessels and the optic disk. The segmented regions were then removed from the original image and the green channel of the RGB image was enhanced. Finally, the preprocessed image was passed through a shallow CNN architecture, where the final classification layer, typically composed of a neural network with the softmax activation function, was replaced with a support vector machine (SVM) classifier. Nonetheless, the segmentation-based techniques required more resources since the system needed to hold two models (the segmentation model and the classification model).

Recently, vision transformers have gained attention in a variety of disease classifications. Gu et al. [41] employed a vision transformer for DR classification. The fine-tuned vision transformer accommodated a residual attention unit at its encoder to enhance the performance. Yao et al. [42] employed Swin Transformer with transfer learning to classify DR images from the Messidor dataset. With extensive hyperparameter tuning, the model achieved 90% accuracy. Dihin et al. [43] also leveraged the Swin Transformer to classify DR images. Regarding binary class classification, the model achieved 97% accuracy.

According to the studies discussed in this section, Table 1 presents a brief overview of the methods, results, and their limitations. The majority of research works have three dominant problems. The issues include low classification performance, high computational complexity, and less diversity in the dataset. Among the three prime issues, this paper addresses the first two, presenting a lightweight DR classifier with high performance. Moreover, we evaluated the proposed model against recent literature using different evaluation metrics on the APTOS and IDRiD datasets.

**Table 1.** Comparison with different methods.

| Paper | Method | Dataset | Accuracy | Limitation |
|---|---|---|---|---|
| Usman et al. [2] | ResNet152, PCA | DR and CSME | 93.67% | Experiment conducted on one dataset |
| Bilal et al. [40] | U-Net, Inception-V3 | Messidor-2, EyePACS-1, and DIARETDB0 | EyePACS-1: 97.92%, Messidor-2: 94.59%, DIARETDB0: 93.52% | Has a relatively high computational cost. |
| Dihin et al. [43] | Swin Transformer | APTOS | 96% (Binary Classification) | No comparison provided and works only on binary classification. |
| Yao et al. [42] | Swin Transformer | Messidor | 98.66% | Experiment limited to one dataset and lacks detailed experimental outcomes. |
| Gu et al. [41] | Vision Transformer | DDR, IDRiD | DDR: 91.54% IDRiD: 87.92% | Quadratic self-attention time. Therefore, resource-consuming. |
| Farag et al. [27] | CNN, DenseNet169 | APTOS | 82% | Low classification performance. |
| Mondal et al. [29] | GAN, DenseNet101, ResNext | APTOS | 86.08% | Low classification performance. |
| Menaouer et al. [30] | VGG with custom CNN | APTOS | 90% | High computation and results are biased toward the majority class. |
| Mustafa et al. [34] | ResNet50, DenseNet121, PCA | Messidor-2, EyePACS, APTOS, DDR | Messidor-2 95.58%, EyePACS: 89.20%, APTOS: 89%, DDR: 76.81% | High computation and lacks detailed comparison with existing methods. |
| Mungloo et al. [31] | ResNet50 | APTOS, Mauritian | APTOS: 82% Mauritian: 79% | Low classification performance |
| Das et al. [32] | EfficientNetB4 | EyePACS | 79.11% | Low classification performance |
| Alahmadi [26] | Custom two-stream CNN | APTOS | 85% | Low classification performance |
| Raiaan et al. [35] | Lightweight CNN architecture | Combined dataset (Messidor-2, IDRiD, and APTOS) | 98.65% | High number of trainable parameters. |
| Proposed | Xception, Knowledge Distillation | APTOS, IDRiD | APTOS: 99.04% IDRiD: 94.17% | Experiment conducted on two datasets only. |

## 3. Methodology

This section can be divided into three main steps. The first step is data preprocessing, which is described in Section 3.1. This process involves improving the quality of the original image. The second step involves constructing a robust teacher model, followed by a student model, which is trained through knowledge distillation. Section 3.2 illustrates this process in detail. Thirdly, we conducted an ablation study to select the best hyperparameters. The study is presented in Section 3.3. Figure 1 presents the overview of the full pro.

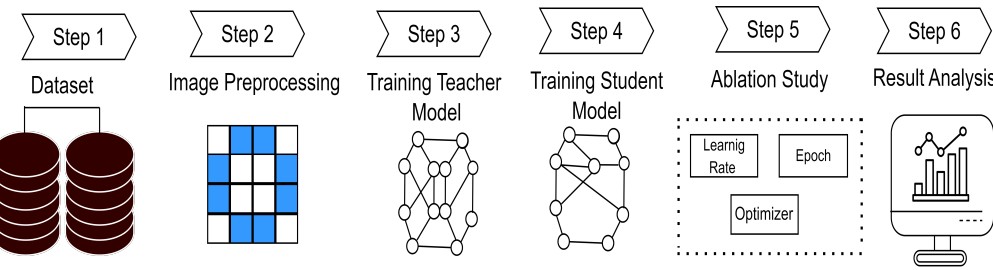

**Figure 1.** Overview of the proposed method.

### 3.1. Data Preprocessing

The datasets used to evaluate the proposed system were the Asia Pacific Teleophthalmology Society (APTOS) 2019 dataset [44] and the Indian Diabetic Retinopathy Image Dataset (IDRiD) 2018 [45]. The APTOS dataset consists of 3662 retinal images of five classes (no DR, mild, moderate, severe, and proliferate) while the IDRiD dataset only consists of 516 DR images. For the imbalanced datasets, the majority of their images are in the no DR class. Table 2 presents the details of the dataset distributions. A sample of the APTOS dataset is presented in Figure 2. Likewise, the IDRiD dataset consists of the same types of images. Image enhancement techniques can vastly influence the classifier's performance and increase visibility [46,47]. Therefore, the proposed system incorporates a series of image-processing algorithms, which can be mainly divided into four stages. A brief illustration of these stages is presented in Figure 3. The algorithms are described below.

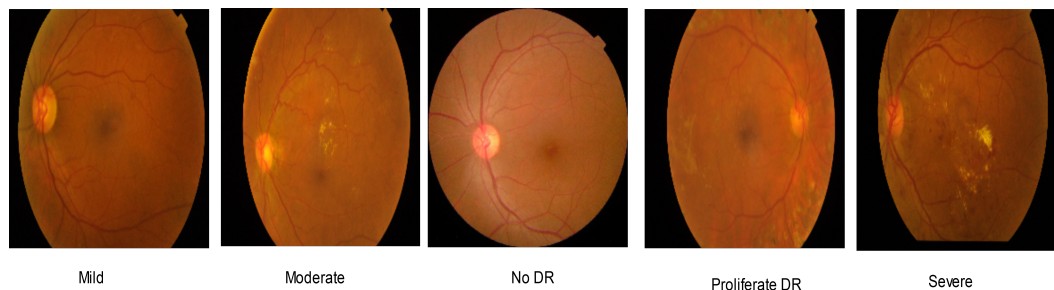

**Figure 2.** Sample of the APTOS dataset.

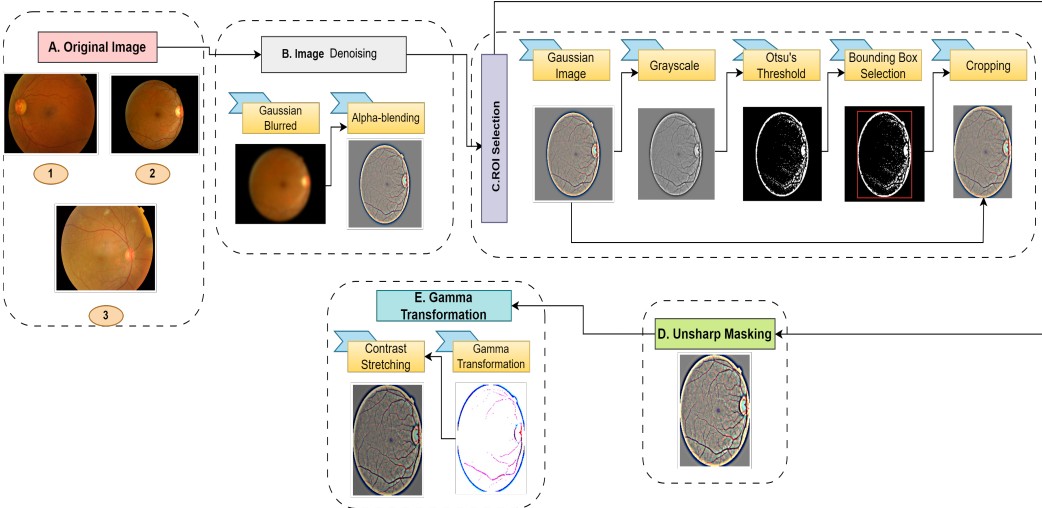

**Figure 3.** Image preprocessing stage.

**Table 2.** Data distribution of the APTOS and IDRiD datasets.

| Grade | APTOS | IDRiD |
|---|---|---|
| Normal | 1805 | 168 |
| Mild | 370 | 25 |
| Moderate | 999 | 168 |
| Serve | 193 | 93 |
| Proliferative | 295 | 62 |
| Total | 3662 | 516 |

### 3.1.1. Image Denoising

Fundus images typically have noise issues. Therefore, the images are passed through a Gaussian kernel to remove the noise [48]. The Gaussian kernel emphasizes the central pixel,

gradually reducing the weights of its surrounding pixels. The priority of the surrounding pixels depends on the standard deviation of the kernel. A larger standard deviation produces a wider Gaussian curve that results in more smoothing. The process of Gaussian filtering can be illustrated with Equations (1) and (2). Here, G is the filter and $\sigma$ is the standard deviation. In the first step, the filter is calculated using Equation (1).

$$G(x,y) = \frac{1}{2\pi\sigma^2}e^{-\frac{x^2+y^2}{2\sigma^2}} \tag{1}$$

Subsequently, the filter is applied to the input image, $I_{\text{input}}$, which produces the smooth image $I_{\text{smoothed}}$.

$$I_{\text{smoothed}}(x,y) = \sum_{i=-k}^{k}\sum_{j=-k}^{k} I_{\text{input}}(x+i, y+j) \cdot G(i,j) \tag{2}$$

In this research, the kernel has a standard deviation of 10 at the *X*-axis and 0 at the *Y*-axis. The kernel produces a smooth image on the horizontal axis and no blurring on the vertical axis. The justification for using only horizontal smoothing on retinal images comes from the fact that many important anatomical features, including blood vessels, lesions, and other important features, predominately have horizontal orientations. In order to enhance these structures and reduce noise that primarily affects horizontal components, smoothing along the *X*-axis is applied.

Finally, the input image and the smooth image are alpha-blended. Alpha blending is an approach used for combining two images by assigning a weight (alpha value) to each pixel, which determines how much the pixel from one image affects the final result. This algorithm is applied to subtract the blurred image from the original image. The refined image has enhanced contrast with attention to the edges and details. Equation (3) illustrates the detailed process of alpha blending. In this equation, $\alpha$ and $\beta$ are two constants used to determine the weights of the original image, $I_{\text{input}}$, and the smooth image, $I_{\text{smoothed}}$. Finally, an offset, C is added to the resulting image to adjust the brightness. For this experiment, the values of $\alpha$, $\beta$, and C are 4, $-4$, and 128, respectively.

$$I_{\text{refined}}(x,y) = \alpha \cdot I_{\text{input}}(x,y) + \beta \cdot I_{\text{smoothed}}(x,y) + C \tag{3}$$

### 3.1.2. ROI Selection

The majority of the images contain a dark background, which is irrelevant to the model. Therefore, the region of interest (ROI) is selected to enhance the model's accuracy. The ROI selection process involves segmenting the image using Otsu's thresholding [49]. For that, the RGB image is first converted into a grayscale image. Subsequently, Otsu's algorithm is applied, which returns a threshold. The process of determining the threshold can be explained via Equation (4). In this equation, in order to determine the threshold, $k_{\text{Otsu}}$, the algorithm iterates over all possible thresholds, $k$. Here, $q_1(k)$ and $\sigma_1^2(k)$ are the probability variances of the background pixels. Similarly, $q_2(k)$ and $\sigma_2^2(k)$ are the probability variances of the foreground pixels, respectively.

$$k_{\text{Otsu}} = \arg\min_{k}\left[q_1(k) \cdot \sigma_1^2(k) + q_2(k) \cdot \sigma_2^2(k)\right] \tag{4}$$

$$I_{\text{output}}(x,y) = \begin{cases} 1, & \text{if } I_{\text{input}}(x,y) > k_{\text{Otsu}} \\ 0, & \text{otherwise} \end{cases} \tag{5}$$

After calculating the global threshold, as explained in Equation (5), the intensity values above the threshold are then marked white and the rest are black. Since the background is dark, the algorithm can easily segment the background from the retinal section. Based on the segmentation, a bounding box is drawn, covering the white region. This box works

as the selection boundary of the ROI. Therefore, the input image is cropped based on the bounding box.

### 3.1.3. Unsharp Masking

In digital image processing, unsharp masking is a popular image improvement technique. By drawing attention to borders and limits, it seeks to improve the appearance of sharpness and fine details in an image [50,51]. This process can be divided into three steps. Firstly, it applies a low-pass filter to the original image to capture the low-frequency components, resulting in a blurred image. Let the original image be $I_{input}$ and $f$ be a low-pass filter. Now, the blurred image, $I_{blurred}$, can be obtained using Equation (6).

$$I_{blurred} = I_{input} * f \qquad (6)$$

In the second stage, the blurred image is subtracted from the original image to obtain the mask, which contains only high-frequency components. The resulting $I_{mask}$ is the element-wise subtraction of pixel values from the original image and the blurred image. Equation (7) presents the mathematical illustration of this stage.

$$I_{mask} = \sum_{i=1}^{x} \sum_{j=1}^{y} I_{input}(i,j) - I_{blurred}(i,j) \qquad (7)$$

Finally, the mask is amplified and added to the original image. Let the amplification value be denoted with $k$, which is 1 for this research. Now the final unsharp mask image is retrieved, performing the operations mentioned in Equations (8) and (9).

$$I_{mask} = I_{mask} \times k \qquad (8)$$

$$I_{output} = \sum_{i=1}^{x} \sum_{j=1}^{y} I_{mask}(i,j) + I_{input}(i,j) \qquad (9)$$

### 3.1.4. Gamma Transformation

Gamma transformation, a nonlinear operation, is commonly used in image processing to modify an image's brightness levels [52]. It involves modifying the intensity distribution of the pixels by applying a power law function to the pixel value. Let us consider $I_{input}$ to be the image before the gamma transformation and $I_{gamma}$ to be the gamma-transformed image. Therefore, the process of gamma transformation can be explained through Equation (10).

$$I_{gamma} = I_{input}^{\gamma} \qquad (10)$$

The exponent of the power law equation, the gamma value, controls the overall brightness and contrast of an image. The image becomes darker when the gamma value is raised as the transformation compresses the lower intensity values while maintaining the higher ones. In contrast, lowering the gamma value widens the range of lower-intensity values and compresses the higher ones, making the image appear brighter. Due to the integration of unsharp masking, some of the less important regions are highlighted, which can have a negative impact on the model. Therefore, the gamma transformation is applied to reduce the brightness. After a thorough analysis, the value of gamma, $-5.047$, is carefully selected for this study. After the transformation, the intensity values are stretched to the original range of 0 to 255. Let the image after gamma transformation be denoted with $I_{gamma}$, the maximum intensity value after gamma transformation with $I_{max}$, and the

minimum value with $I_{\min}$. Therefore, the image after contrast stretching, $I_{\text{stretched}}$, can be produced using Equation (11).

$$I_{\text{stretched}}(x, y) = \frac{255}{I_{\max} - I_{\min}} \cdot (I_{\text{gamma}}(x, y) - I_{\min}) \tag{11}$$

Finally, the images are resized to $224 \times 224$ to fit into the pre-trained classifier's input dimension. Figure 4 presents the changes in the image after each stage. Although some images are visually less appealing to the human eye, in the context of the full image, the blood vessels and exudates are more emphasized than less important regions. The system, however, does not integrate any data augmentation techniques.

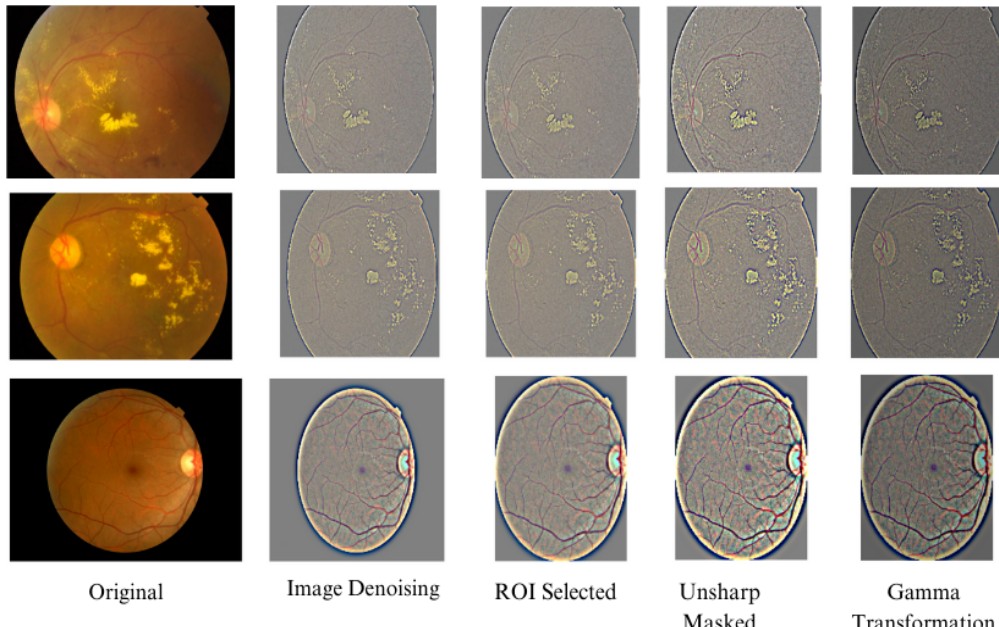

Original  Image Denoising  ROI Selected  Unsharp Masked  Gamma Transformation

**Figure 4.** Change in retinal images after a series of preprocessing.

### 3.2. Model Construction

In this stage, the teacher model is constructed first, followed by the student model. Finally, the student model is trained through the distillation loss and cross-entropy. The subsequent sections present these steps in detail.

### 3.2.1. Teacher Model

The teacher model employed in our system consists of two high-performing image classifiers: Swin Transformer and ResNet152V2. The Swin Transformer was developed as an advancement to the Vision Transformer (ViT). The major issue faced in vision transformers involves the quadratic time complexity associated with computing self-attention since each patch needs to look into every other patch to compute self-attention. Swin Transformer introduced a solution to this problem by integrating shifted windows and hierarchical transformer architecture. In the shifted window mechanism, images are divided into fixed-size windows. To compute self-attention, the patches in a particular window only focus on the other patches of that window, which dramatically reduces resource consumption. The shifted window mechanism also allows the architecture to gain a higher contextual understanding beyond a mere window-centric focus. The hierarchical block architecture, on the other hand, combines features across different scales. This mechanism significantly improves the classification performance, making Swin Transformer one of the most powerful image classifiers available [53].

ResNet152V2 is a CNN architecture that is based on ResNet [54]. The backbone of this model is the residual connection. The skip connections used in residual blocks

allow the network to retain information extracted from previous blocks and propagate throughout the network. This also erases the vanishing gradient problem, which allows large model construction.

As presented in Figure 5, the proposed teacher model is a fusion model made of Swin Transformer and ResNet152V2. The features extracted from both networks follow a batch normalization for smoother convergence. Since ResNet152V2 returns two-dimensional features, global average pooling (GAP) is employed for converting the 2D feature maps to the 1D feature vector. GAP was found to be a more efficient approach for this task than flattening. Finally, features from both networks are fused and passed to a dropout layer, where 50% of features are randomly dropped to regularize the model. A 50% dropout rate in the teacher model ensures that the model does not overfit in the highly imbalanced dataset. This learning is then transferred to the student model, eliminating the need for additional regularization. The dropout layer then follows a dense layer, which is composed of 256 neurons, and is activated with the SeLU activation function. SeLU is a simple yet effective activation function developed to address the dying ReLu problem [55]. The final layer consists of two and five neurons, with softmax activation for binary and multi-class classifications, respectively. The optimizer chosen for this task is Adamax, which provides smoother and faster convergence [56].

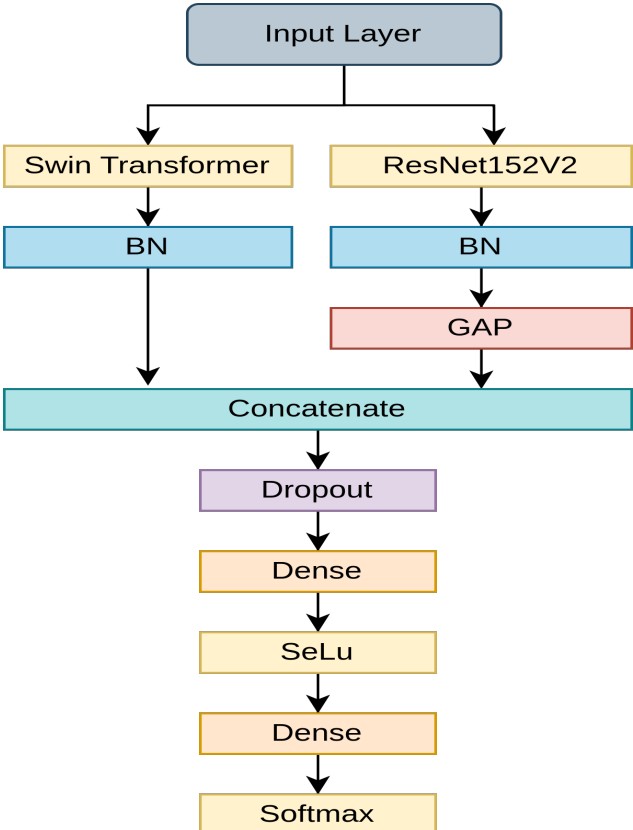

**Figure 5.** The proposed teacher model.

### 3.2.2. Student Model

The student model selected for DR detection is made of the Xception [57] backbone. Xception is a high-performing CNN architecture that replaces the inception module of the Inception network with a modified depthwise separable convolution (DWSC). DWSC is a lightweight convolution module that is composed of depthwise convolution, followed by pointwise convolution. DWSC has been widely used in MobileNets due to its efficiency in reducing computational complexity, without compromising performance [58]. In the modified DWSC, integrated into Xception, however, pointwise convolution is applied first, which then follows depthwise convolution. Due to the lightweight nature and high

performance of Xception, it is an ideal fit for the student model. Furthermore, many studies have found that the model performs exceptionally well in DR classification [59,60]. As shown in Figure 6, the Xception backbone is fine-tuned with a modified CBAM block.

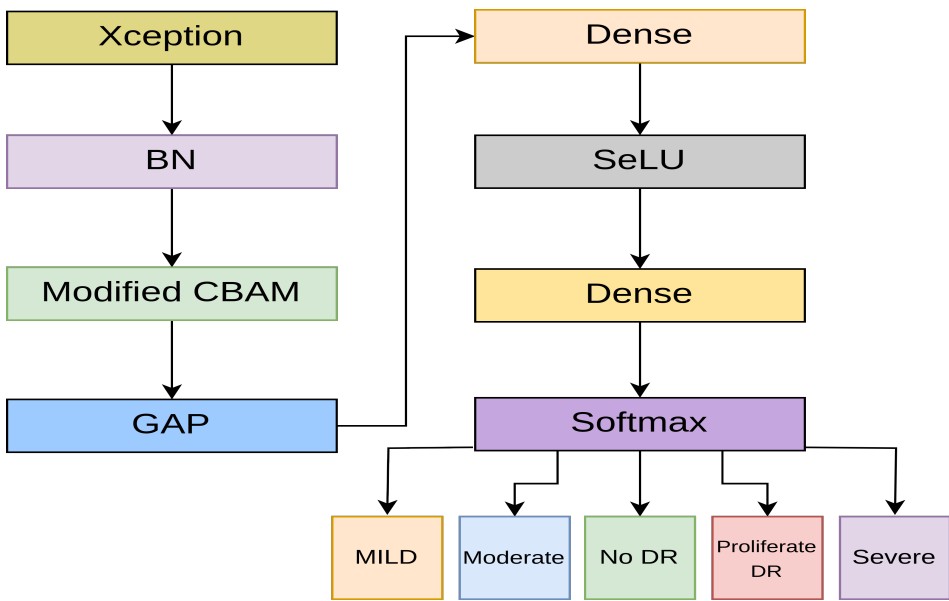

**Figure 6.** The proposed student model.

CBAM is a widely used attention module that encompasses both channel attention and spatial attention [28]. Therefore, integrating CBAM into the architecture boosts the performance. As presented in Figure 7, the spatial attention module of CBAM usually consists of a convolutional layer with a kernel size of 7. Due to the involvement of a large kernel, CBAM requires large computational power. Therefore, the modified CBAM integrates a depthwise separable convolution (DWSC) to reduce the future size by a factor of 2, leading to fewer computations. Moreover, the final convolutional layer leverages dilated convolution with a dilation rate of 2. The dilated convolution can capture large-scale features without consuming high resources.

### 3.2.3. Training the Student Model with Knowledge Distillation

Knowledge distillation is a cutting-edge method used to transfer information from a more complicated teacher model to a simpler student model. Generally, larger models have higher classification accuracy than lighter ones [61]. By constricting a larger model with higher performance and a light model with lower performance, we force the lighter model to imitate the behavior of the larger one, while focusing on cross-entropy loss to improve its classification performance. This process is generally referred to as knowledge distillation. In knowledge distillation, the larger model is known as the teacher model, and the lighter model is the student model. In this experiment, a hybrid ResNet152V2 and Swin Transformer architecture is employed as the teacher model. The main objective is to transfer the teacher model's high-level decision boundaries and feature representations to the student model, Xception, so that it can imitate them and gain access to its discriminative power.

As presented in Figure 8, both the teacher and student models receive the same input data during training, and the student model is tuned to produce predictions that are both accurate and consistent with the softer, more nuanced outputs of the teacher model. This alignment is accomplished by combining supervised loss, which guarantees accurate class predictions, and distillation loss, which incentivizes the student to match the probability distributions of the teacher, typically by using a temperature parameter to regulate the output softness of the teacher. Knowledge distillation produces a more compact model with competitive performance by successfully transferring the teacher model's comprehensive understanding and generalization abilities to the student. To train both models, we used

the categorical cross-entropy loss function and Adam optimizer. The training process lasted for 100 epochs. The hyperparameter temperature value is 10, making the probability distribution of the teacher model more soft.

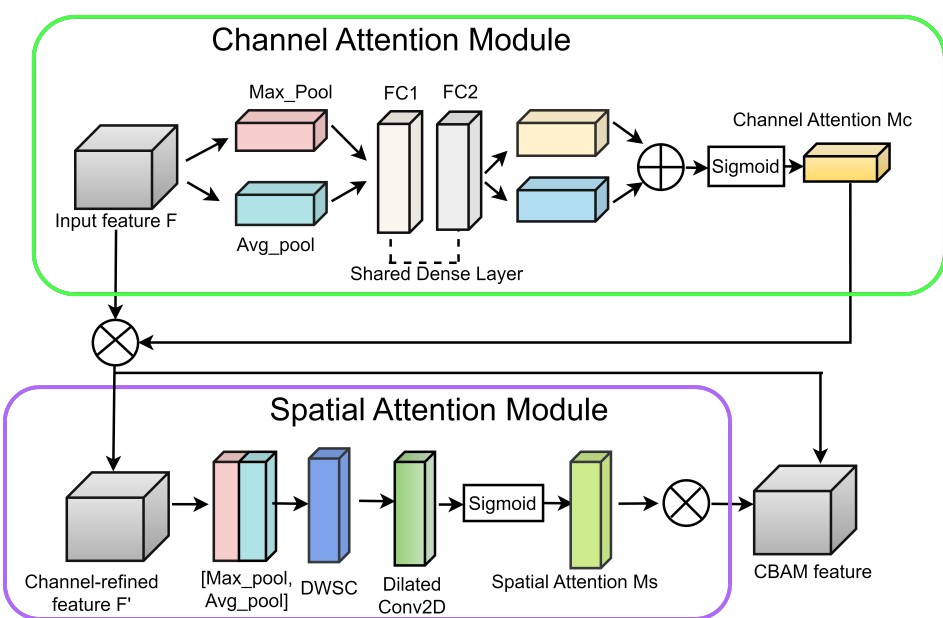

**Figure 7.** The modified CBAM block.

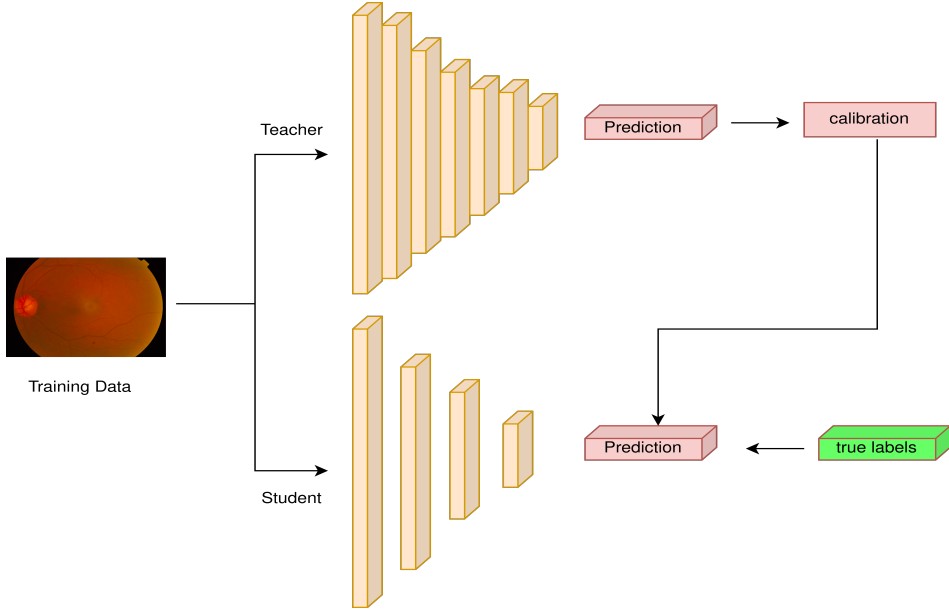

**Figure 8.** Knowledge distillation process overview.

### 3.3. Ablation Study

We conducted an ablation study on the student model to find the optimal values of the hyperparameters. Table 3 contains the accuracy of the APTOS and IDRiD datasets with respect to the changes in the hyperparameters. The results show that the Adamax optimizer outperforms Adam and RMSProp by a small margin on the APTOS dataset. On the IDRiD dataset, however, Adam outperforms Adamax by a slight margin. In this conflicting situation, we selected Adamax since it provides a smoother convergence compared to Adam [62]. The best performance is provided by a learning rate of 0.001 on both datasets. Similarly, 100 epochs produce the best performance. Therefore, the best configuration is selected as an Adamax optimizer with a 0.001 learning rate and 100 epochs of training.

**Table 3.** Investigation of the hyperparameters on the student model.

| Hyperparameter | Value | Accuracy on APTOS | Accuracy on IDRiD |
|---|---|---|---|
| Optimizer | Adam | 0.9843 | 0.9270 |
| | Adamax | 0.9861 | 0.9236 |
| | RMSProp | 0.9702 | 0.9061 |
| Learning Rate | 0.1 | 0.9636 | 0.9346 |
| | 0.005 | 0.9690 | 0.9308 |
| | 0.001 | 0.9781 | 0.9383 |
| Epoch | 20 | 0.8256 | 0.8642 |
| | 50 | 0.9411 | 0.9259 |
| | 100 | 0.9904 | 0.9417 |

## 4. Results

This section presents the results obtained from the experiment along with a thorough comparison with recent literature.

### 4.1. Experimental Setup

The experiment is conducted on Kaggle. The programming language used for this experiment is Python (version 3.7.6). Six Python libraries are leveraged to ease up the experiment: TensorFlow, matplotlib, sklearn, pandas, NumPy, and OS.

### 4.2. Result Analysis

The evaluation process employs various evaluation metrics, including accuracy, precision, recall, F1 score, and the Matthews correlation coefficient (MCC). Among these metrics, MCC is the most informative one since it accommodates all the coordinates [63]. The metrics can be calculated using the following equations.

$$Accuracy = \frac{TP + TN}{TP + PP + TN + FN} \tag{12}$$

$$Precision = \frac{TP}{TP + FP} \tag{13}$$

$$Recall = \frac{TP}{TP + FN} \tag{14}$$

$$F1 - score = \frac{2 \times Precision \times Recall}{Precision + Recall} \tag{15}$$

$$MCC = \frac{TP \times TN - FP \times FN}{\sqrt{(TP + FP)(TP + FN)(TN + FP)(TN + FN)}} \tag{16}$$

For further evaluation, we present the confusion matrix and the receiver operating characteristic (ROC) curve. Various models have been tested for the teacher model. Table 4 presents a short description of the teacher models. While the proposed teacher model has over 254 million parameters, its only responsibility is to train the student model. Given that the student model is intended for use in real-world applications, it is essential that its architecture be lightweight, emphasizing efficiency and practicality in operational contexts. Tables 5 and 6 present a comparison of different teacher models on the APTOS and IDRiD datasets, respectively. Figures 9 and 10 present graphical overviews of the comparison on the APTOS and IDRiD datasets, respectively. Among the eight models, the proposed fusion model based on Swin Transformer and ResNet152V2 performs the best. Traditionally, fusion models perform better than single-model classifiers. Moreover, the suggested teacher model incorporates two different feature extractors (CNN and transformer), which have totally different methods for feature extraction. Therefore, the extracted features only have

a few overlaps. Additionally, Swin Transformer is a recent image classifier that outperforms traditional classifiers. Therefore, with small fine-tuning, the model is able to achieve high performance.

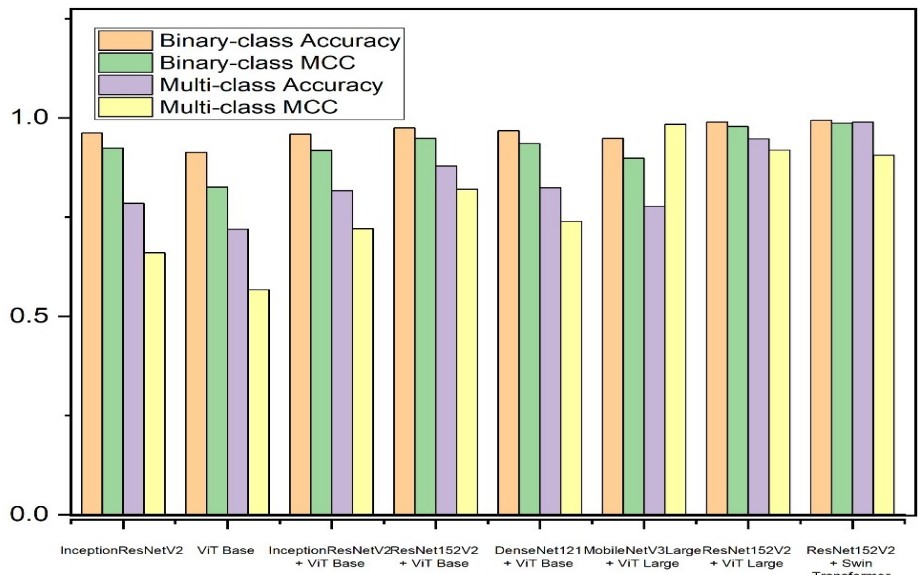

**Figure 9.** The teacher model performance comparison on the APTOS dataset.

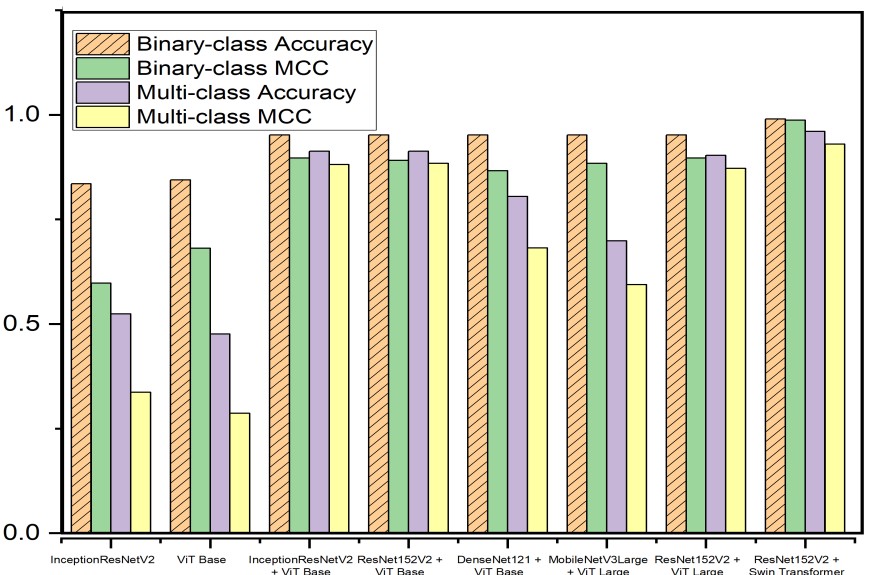

**Figure 10.** The teacher model performance comparison on the IDRiD dataset.

**Table 4.** Teacher model information and parameters.

| Model | Size (MB) | Total Parameters | Trainable Parameters |
|---|---|---|---|
| Inception-ResNet-v2 | 216.5 | 54.35 M | 0.4 M |
| ViT Base | 350.1 | 87.46 M | 0.3 M |
| Inception-ResNet-v2 + ViT Base | 568.7 | 141.81 M | 0.9 M |
| ResNet152V2 + ViT Base | 584.4 | 145.81 M | 0.6 M |
| DenseNet121 + ViT Base | 379 | 94.51 M | 0.7 M |
| MobileNetV3Large + ViT Large | 1010.7 | 308.52 M | 0.7 M |
| ResNet152V2 + ViT Large | 1230.9 | 363.86 M | 1.2 M |
| ResNet152V2 + Swin Transformer | 1460.3 | 254.26 M | 1.4 M |

**Table 5.** Comparison with different teacher methods on the APTOS dataset.

| Model | Binary Class Accuracy | Binary Class MCC | Multi-Class Accuracy | Multi-Class MCC |
|---|---|---|---|---|
| Inception-ResNet-v2 | 0.9618 | 0.9236 | 0.7844 | 0.6601 |
| ViT Base | 0.9127 | 0.8250 | 0.7190 | 0.5666 |
| Inception-ResNet-v2 + ViT Base | 0.9591 | 0.9180 | 0.8158 | 0.7207 |
| ResNet152V2 + ViT Base | 0.9741 | 0.9484 | 0.8786 | 0.8199 |
| DenseNet121 + ViT Base | 0.9673 | 0.9352 | 0.8240 | 0.7387 |
| MobileNetV3Large + ViT Large | 0.9482 | 0.8983 | 0.7763 | 0.9834 |
| ResNet152V2 + ViT Large | 0.9891 | 0.9781 | 0.9468 | 0.9186 |
| ResNet152V2 + Swin Transformer | 0.9932 | 0.9864 | 0.9891 | 0.9061 |

**Table 6.** Comparison of different teacher models on the IDRiD dataset.

| Model | Binary Class Accuracy | Binary Class MCC | Multi-Class Accuracy | Multi-Class MCC |
|---|---|---|---|---|
| Inception-ResNet-v2 | 0.8350 | 0.5976 | 0.5243 | 0.3367 |
| ViT Base | 0.8447 | 0.6812 | 0.4757 | 0.2867 |
| Inception-ResNet-v2 + ViT Base | 0.9515 | 0.8970 | 0.9126 | 0.8809 |
| ResNet152V2 + ViT Base | 0.9515 | 0.8913 | 0.9126 | 0.8838 |
| DenseNet121 + ViT Base | 0.9519 | 0.8664 | 0.8046 | 0.6814 |
| MobileNetV3Large + ViT Large | 0.9515 | 0.8838 | 0.6990 | 0.5943 |
| ResNet152V2 + ViT Large | 0.9515 | 0.8970 | 0.9029 | 0.8721 |
| ResNet152V2 + Swin Transformer | 0.9903 | 0.9873 | 0.9601 | 0.9301 |

For the student model, nine lightweight state-of-the-art image classifiers were considered. Table 7 holds the details of the classifiers. Among them, Xception performed the best and EfficientNetV2B0 performed the worst. Three versions of the EfficientNet models were considered. EfficientNet architecture achieved high performance in various image classification tasks; however, the models failed to achieve acceptable results in DR classification. Although Xception did not have the fewest parameters and was relatively larger in size, it outperformed all other models, taking into account the comparative analysis, with a remarkable accuracy of 86.23% on the APTOS dataset, with an acceptable size. With knowledge distillation, the accuracy increased to 87.99%. The multi-class accuracy of the student model on the two datasets is presented in Figure 11. Tables 8 and 9 present the comparisons of different student modes with and without knowledge distillation, respectively, on the APTOS dataset. Likewise, Table 10 presents a comparison on the IDRiD dataset. The knowledge distillation process also improved its accuracy by 2.38%.

**Table 7.** Student model information and parameters.

| Model | Size (MB) | Total Parameters | Trainable Parameters |
|---|---|---|---|
| NASNetMobile | 19.4 | 4.5 M | 0.2 M |
| DenseNet201 | 73.7 | 18.8 M | 0.4 M |
| MobileNetV2 | 10.4 | 2.6 M | 0.3 M |
| MobileNet | 13.6 | 3.5 M | 0.2 M |
| ResNet50 | 92.5 | 24.2 M | 0.5 M |
| EfficientNetV2B0 | 24.6 | 6.3 M | 0.3 M |
| EfficientNetB7 | 248.7 | 64.7 M | 0.6 M |
| EfficientNetB4 | 70.2 | 18.2 M | 0.4 M |
| Xception | 82 | 21.4 M | 0.5 M |

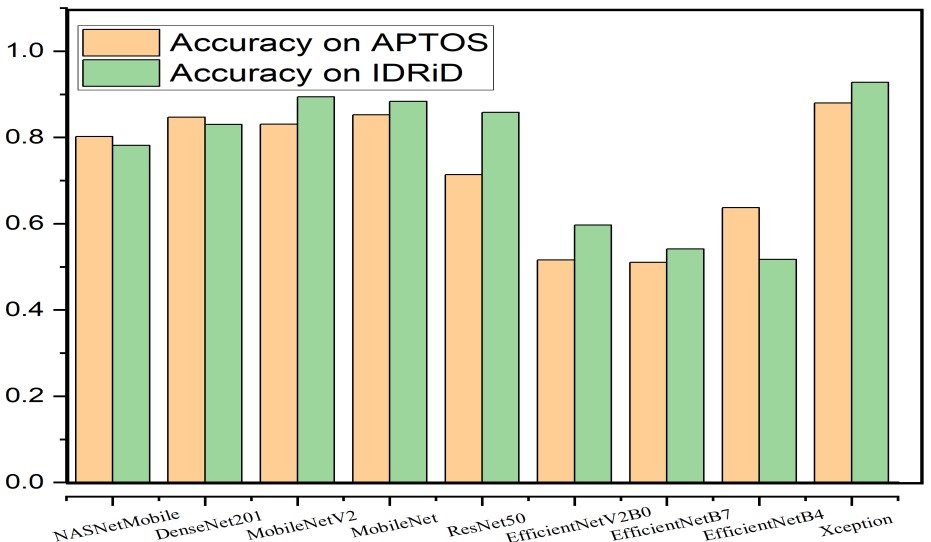

**Figure 11.** Multi-class accuracy of different student models on the APTOS and IDRiD datasets.

**Table 8.** Comparison of base models on the APTOS and IDRiD datasets.

| Model | Multi-Class Accuracy (APTOS) | Multi-Class MCC (APTOS) | Multi-Class Accuracy (IDRiD) | Multi-Class MCC (IDRiD) |
|---|---|---|---|---|
| NASNetMobile | 0.8158 | 0.7134 | 0.7723 | 0.6363 |
| DenseNet201 | 0.8322 | 0.7523 | 0.8208 | 0.7190 |
| MobileNetV2 | 0.8022 | 0.6911 | 0.7903 | 0.5047 |
| MobileNet | 0.8336 | 0.7450 | 0.7894 | 0.6809 |
| ResNet50 | 0.7531 | 0.6152 | 0.6239 | 0.4216 |
| EfficientNetV2B0 | 0.4748 | 0.0000 | 0.5170 | 0.4003 |
| EfficientNetB7 | 0.4911 | 0.0000 | 0.4658 | 0.3583 |
| EfficientNetB4 | 0.6112 | 0.3917 | 0.5062 | 0.6419 |
| Xception | 0.8623 | 0.8194 | 0.9041 | 0.8850 |

**Table 9.** Comparison of different student models on the APTOS dataset.

| Model | Binary Class Accuracy | Binary Class MCC | Multi-Class Accuracy | Multi-Class MCC |
|---|---|---|---|---|
| NASNetMobile | 0.9591 | 0.9180 | 0.8022 | 0.7060 |
| DenseNet201 | 0.9673 | 0.9346 | 0.8472 | 0.7738 |
| MobileNetV2 | 0.9645 | 0.9298 | 0.8308 | 0.7408 |
| MobileNet | 0.9714 | 0.9429 | 0.8527 | 0.7747 |
| ResNet50 | 0.9372 | 0.8748 | 0.7135 | 0.5720 |
| EfficientNetV2B0 | 0.6985 | 0.4643 | 0.5157 | 0.0000 |
| EfficientNetB7 | 0.8226 | 0.6510 | 0.5102 | 0.0000 |
| EfficientNetB4 | 0.8663 | 0.7312 | 0.6371 | 0.4530 |
| Xception | 0.9673 | 0.9345 | 0.8799 | 0.8194 |

**Table 10.** Comparison of different student models on the IDRiD dataset.

| Model | Binary Class Accuracy | Binary Class MCC | Multi-Class Accuracy | Multi-Class MCC |
|---|---|---|---|---|
| NASNetMobile | 0.9130 | 0.7980 | 0.7819 | 0.6696 |
| DenseNet201 | 0.9252 | 0.6341 | 0.8304 | 0.7507 |
| MobileNetV2 | 0.9530 | 0.8490 | 0.8941 | 0.8290 |
| MobileNet | 0.9158 | 0.4480 | 0.8836 | 0.6973 |
| ResNet50 | 0.8224 | 0.6020 | 0.8579 | 0.6138 |
| EfficientNetV2B0 | 0.7757 | 0.4583 | 0.5970 | 0.4801 |
| EfficientNetB7 | 0.7280 | 0.4561 | 0.5412 | 0.4397 |
| EfficientNetB4 | 0.7273 | 0.1267 | 0.5170 | 0.4003 |
| Xception | 0.9623 | 0.9094 | 0.9279 | 0.892 |

Since Xception performed the best among the nine classifiers, it was further modified by integrating a modified CBAM block. Although the integration of the CBAM module increased its size by 20 megabytes, it ameliorated its binary class accuracy by 3.27% and multi-class accuracy by 11.04% on the APTOS dataset. Moreover, there was an enhancement of 1.82% in binary class accuracy and 1.38% in multi-class accuracy on the IDRiD dataset. This significant performance improvement can be attributed to the modified CBAM block's improved adaptability, which improves feature prioritization and selection. Given that the original Xception model, while powerful, may include extraneous features from its pre-training on ImageNet, the CBAM module effectively focuses on the salient features that are specifically pertinent to diabetic retinopathy. This feature improvement leads to a significant increase in classification accuracy, demonstrating the effectiveness of the CBAM module in enhancing the model's representational capabilities. On the APTOS dataset, the proposed student model thereby achieves 0.99045, 0.9907, 0.9905, 0.9903, and 0.9856 in the accuracy, precision, recall, F1 score, and MCC, respectively, on multi-class classification, and a full 1.0 in all metrics for the binary class classification. The confusion matrix, presented in Figure 12, shows that only four moderate images are misclassified as mild and severe. Other than moderate, no other types have more than two misclassifications. Moreover, the confusion matrix emphasizes that the impact of class imbalance on the results is insignificant, as there are no extra predictions in the majority class (no DR). The ROC curve represents the model's ability to discriminate true positive instances from false positive instances. The proposed model's ROC curve, presented in Figure 13, shows a perfect area under the curve with the highest value of 1.0. These results illustrate the model's discriminate ability in classifying DR images. Similarly, on the IDRiD dataset, it achieves 0.9805, 0.9811, 0.9805, 0.9804, and 0.9576 in accuracy, precision, recall, F1 score, and MCC, respectively, on multi-class classification, and 0.9417, 0.9429, 0.9417, 0.9415, and 0.9217 in accuracy, precision, recall, F1 score, and MCC, respectively, on binary classification. The confusion matrix, presented in Figure 14, shows that—similar to the APTOS dataset—the model also faces some challenges in classifying two mild-type DR images in the IDRiD dataset. Other than that, one moderate image and one severe DR image are misclassified as mild and proliferate DR, respectively. The ROC curve presented in Figure 15 also certifies its discriminative ability in differentiating true positive instances from false positive ones with an average ROC of 0.99.

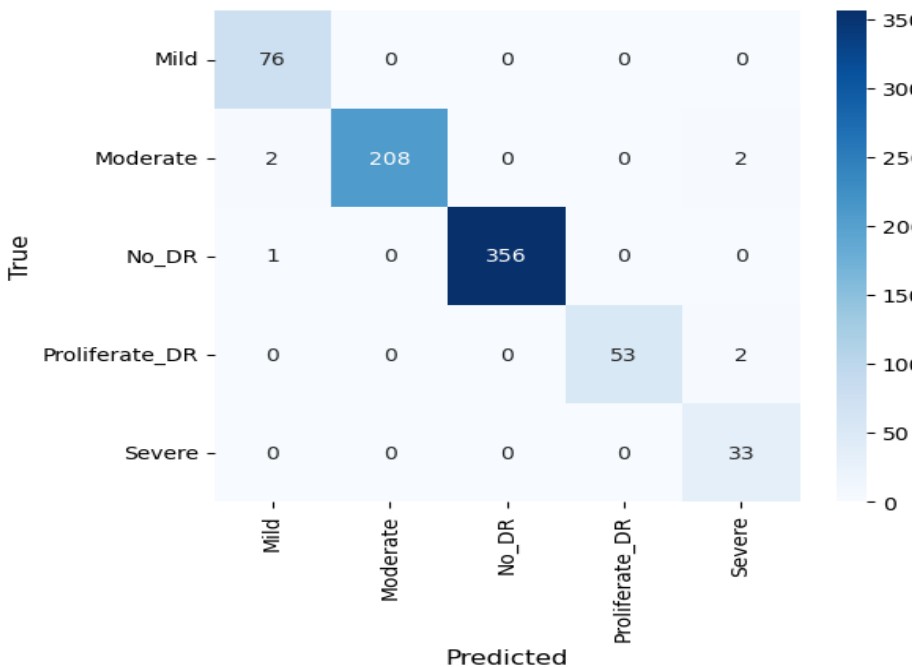

**Figure 12.** Concision matrix of the student model on the APTOS dataset.

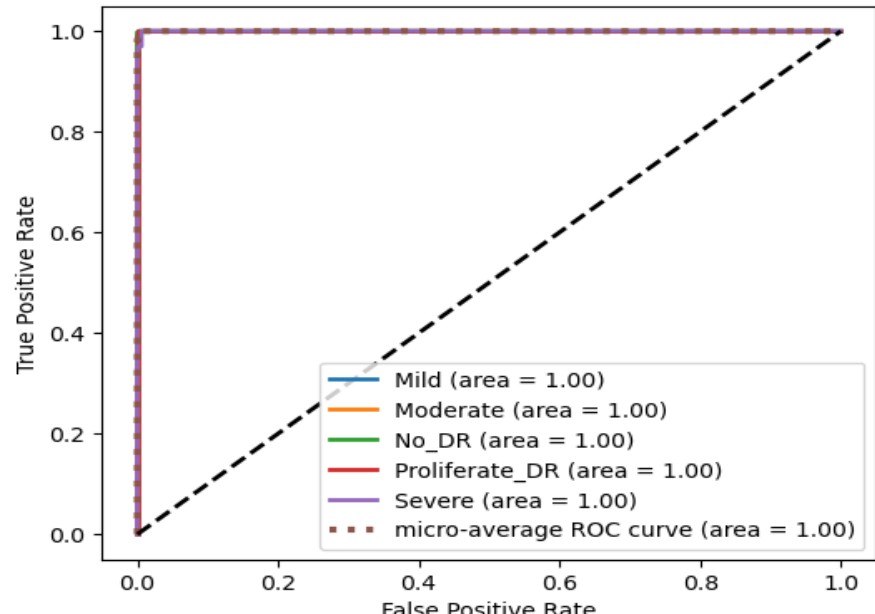

**Figure 13.** ROC curve of the student model on the APTOS dataset.

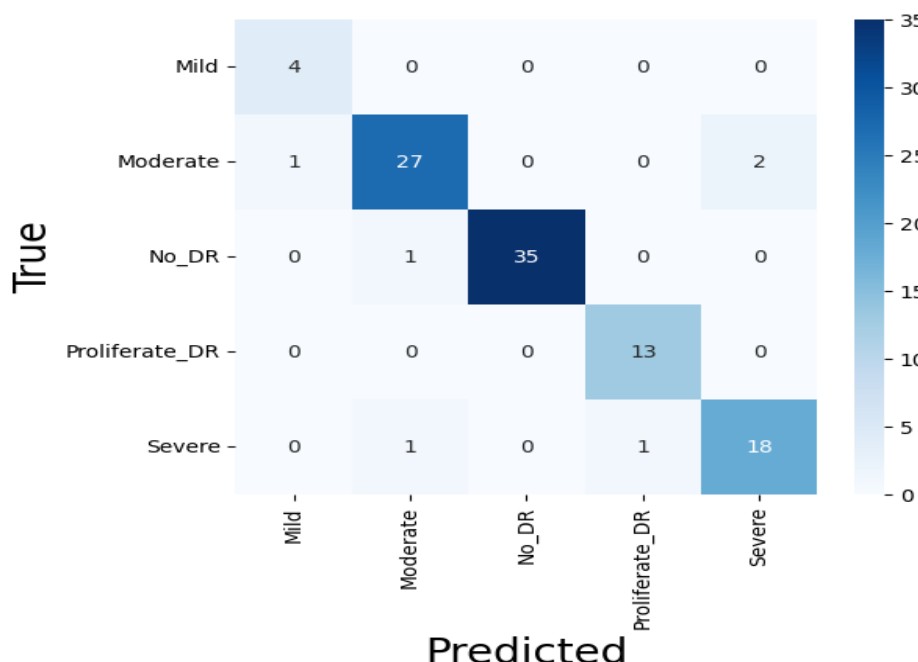

**Figure 14.** Concision matrix of the student model on the IDRiD dataset.

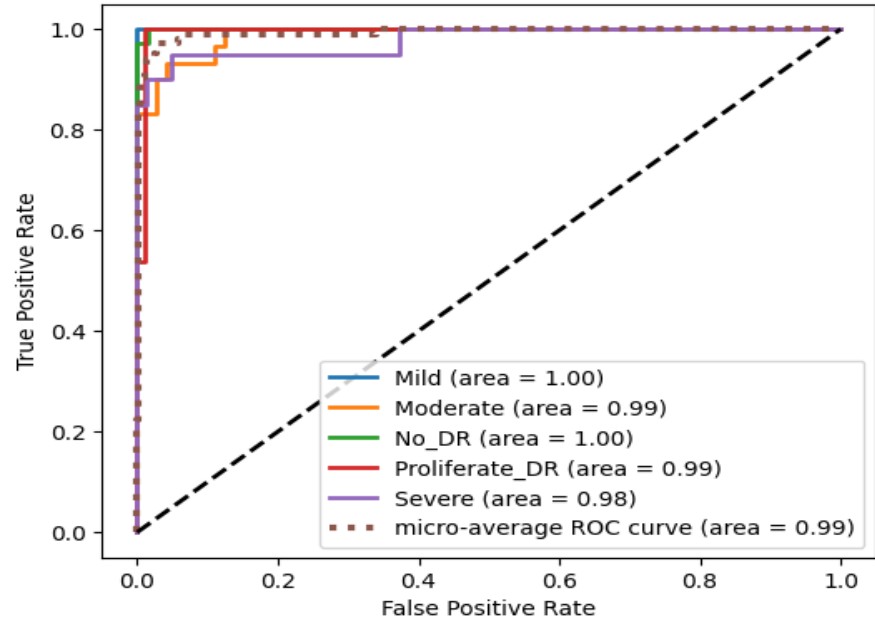

**Figure 15.** ROC curve of the student model on the IDRiD dataset.

### 4.3. Comparison with Existing Works

The majority of the works have leveraged various pre-trained CNN classifiers. This research, however, employs a CNN and transformer-based fusion architecture to improve the classification performance of a lightweight image classifier. Moreover, unlike many studies, this study integrates various image preprocessing tasks to improve the image quality where crucial regions in retinal images are highlighted. With an extensive ablation study, we figured out the best configurations that have made our research stand out. A comparison with existing works on the APTOS dataset is presented in Table 11, showing the superiority of our model in classifying a wide range of DR. On the IDRiD dataset, however, there are only a few research works available. Table 12 presents a comparison with some of the existing methods on the datasets. The table illustrates that the model achieves superior

performance when dealing with small datasets. With the high classification performance consuming low resources, this model can be a vital solution for real-life DR classification, especially for moderate devices.

**Table 11.** Comparison with existing literature on the APTOS dataset.

| Architecture | Accuracy | Precision | Recall | F1 Score |
|---|---|---|---|---|
| DenseNet201 [64] | 0.86 | - | - | 0.7251 |
| ResNet50 [31] | 0.82 | - | - | - |
| Texture attention, spatial attention [26] | 0.851 | - | 0.903 | 0.984 |
| DenseNet121 [65] | 0.9730 | - | - | - |
| Squeeze excitation-based dense network [65] | 0.86 | 0.77 | 0.7 | 0.73 |
| VGG16 and capsule network [66] | 0.99 | - | - | - |
| Semi-supervised auto-encoder graph network [67] | 0.944 | - | 0.84 | - |
| Improved ResNet-50 [68] | 0.9825 | 0.9986 | 0.9584 | 0.99856 |
| DenseNet-121 [69] | 0.9836 | 0.98 | 0.98 | 0.98 |
| Parallel CNN [10] | 0.9727 | 0.96 | 0.95 | 0.95 |
| Proposed | 0.9904 | 0.9906 | 0.9904 | 0.9903 |

**Table 12.** Comparison with existing literature on the IDRiD dataset.

| Architecture | Accuracy | Precision | Recall | F1 Score |
|---|---|---|---|---|
| ResNet [70] | 0.9029 | 0.8875 | 0.9689 | - |
| Cross-disease attention [71] | 0.926 | - | - | - |
| KNN [72] | 0.94 | - | - | - |
| GNN [73] | 0.773 | - | - | - |
| Proposed | 0.9417 | 0.9429 | 0.9417 | 0.9415 |

*4.4. Discussion*

This research presents a robust DR image classifier that achieves state-of-the-art performance by employing a lightweight image classifier and knowledge distillation concept. The teacher model employed in this system is a fusion model composed of ResNet152V2 and Swin Transformer. Although the teacher model has a relatively large size, the model is only leveraged to train the student model. The Xception classifier is used as the student model that achieves accuracies of 86.23% and 90.41% with only cross-entropy loss on the APTOS and IDRiD datasets, respectively. The model's performance improves by an average of 2.07% by combining distillation loss. The performance further improves by 11.05% with the integration of modified CBAM on the APTOS dataset and by 1.38% on the IDRiD dataset. The student model achieves an overall accuracy of 99.04%, which outperforms the teacher model by 0.13% on the APTOS dataset. Nonetheless, on the IDRiD dataset, the student model fails to outperform the teacher model, although the student model achieves a noteworthy performance. Three major reasons can be attributed to the exceptional performance of the student model. Firstly, the integration of the modified CBAM block plays a crucial role in achieving high accuracy. Since the pre-trained classifiers are trained on ImageNet, all the extracted features are not fully relevant to the classification process. Employing an attention module helps boost performance. Secondly, the temperature hyperparameter employed in the training process of the student model is set to 10, which is a higher value. This allows the student model to learn from the distillation loss while primarily focusing on the cross-entropy loss. Finally, the extensive ablation study with a series of preprocessing techniques helped the model achieve a noteworthy performance.

## 5. Conclusions

In this paper, we present a lightweight DR image classifier that achieves high classification performance. The proposed solution involves four stages of image preprocessing: image denoising, ROI extraction, unsharp making, and gamma transformation. The teacher model is developed by fusing Swin Transformer and ResNet152V2. The student model, on the other hand, is constructed using Xception and modified CBAM, which is trained through both supervised loss (cross-entropy loss) and distillation loss. A comparison with existing studies demonstrates the model's superior performance. However, the research work has certain limitations that can be addressed in future endeavors. Firstly, the model can be further compressed employing various model compression tools, like ONNX, which sometimes results in degraded performance. A study could be conducted to explore additional optimization techniques without compromising performance. Moreover, this study only considers one dataset for evaluating the model's performance. In the future, additional datasets could be employed to assess the model's generalization ability.

**Author Contributions:** Conceptualization, N.I.; methodology, N.I.; software, N.I., S.S.; validation, E.H., M.M.H.J.; formal analysis, M.M.I.; investigation, M.M.I.; data curation, N.I.; writing—original draft preparation, N.I., M.M.H.J., E.H., S.S., A.R.; writing—review and editing, M.M.I.; visualization, E.H.; supervision, M.M.I.; project administration, M.M.I.; funding acquisition, M.M.I. All authors have read and agreed to the published version of the manuscript.

**Funding:** This project is funded by the United International University, grant number IAR-2023-Pub-050.

**Institutional Review Board Statement:** Not applicable.

**Informed Consent Statement:** Not applicable.

**Data Availability Statement:** Publicly available datasets are used for this study. APTOS dataset at https://www.kaggle.com/c/aptos2019-blindness-detection/overview, accessed on 19 October 2023 and IDRiD at https://ieee-dataport.org/open-access/indian-diabetic-retinopathy-image-dataset-idrid (accessed on 19 October 2023).

**Conflicts of Interest:** The authors declare no conflict of interest.

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
