# Peer review of "Toward Lightweight Diabetic Retinopathy Classification: A Knowledge Distillation Approach for Resource-Constrained Settings"

_applsci, doi:10.3390/app132212397_

Round 1

Reviewer 1 Report

The author proposed a lightweight model for diabetic retinopathy classification. The algorithm has achieved higher accuracy in terms of classification. The authors used exiting methods and obtained good results. However, there are some comments and suggestions that should be adjusted in the revised version. The comments/suggestions are given below:

1-      Overall, the abstract provides a concise overview of the research, its methodology, and the significant outcomes. It piques interest and indicates a thorough research study. Just ensure that the claims, especially regarding accuracy, are backed by robust methodology and validation techniques. However, the authors should add that how many datasets, metrics, and state of the art techniques were used for validation.

2-      The authors should remove the first point from the contribution’s points. Besides, in line 100, the authors note that their research employs various preprocessing techniques to enhance image quality. Moreover, Swin Transformer and ResNet152V2 are also mentioned in the contributions, but the abstract does not elaborate on these techniques. It would be beneficial if the authors revised the abstract to align it more closely with the research's content.

3-      Overall, the paper is well-written and makes valuable contributions to lane detection. However, I would like to recommend additional citations to further substantiate your claims and provide a more comprehensive literature review. Specifically, I noticed that you can add little bit about the preprocessing techniques for low-light image enhancement would benefit from citing the work such as:

[DOI: 10.19823/j.cnki.1007-1202.2021.0051]

DOI: 10.3390/sym11040574

DOI: 10.1109/ACCESS.2020.3001206

DOI: 10.1007/s00371-020-01838-0

DOI: 10.19823/j.cnki.1007-1202.2021.0051

DOI: 10.1117/1.JEI.31.4.041213

DOI: 10.1109/JSYST.2023.3262593, which is a seminal paper that offers key insights into low-light image enhancement. Including this reference would not only strengthen your theoretical framework but also give the reader a more complete understanding of the subject matter.

4-      The authors should draw another diagram that shows the overall flow of the proposed model. I have not found it. The authors only added diagrams for individual modules.

5-      In line 412, the authors claimed that we conducted extensive ablation study…. What does it mean? an ablation study can be conducted to check the effect of depended and independent model. The authors should add an ablation study for the proposed model.

English quality is fine. 

Author Response

Please find the review response in the file attached herewith.

Reviewer 2 Report

The paper presents a knowledge distillation-based approach, producing a lightweight student model with a small size of 102 megabytes by combining Xception, a state-of-the-art image classifier, with a customized Convolutional Block Attention Module (CBAM). The model performed exceptionally well

during evaluation on the APTOS dataset, achieving 100% accuracy in binary classification and 99.04% accuracy in multiclass classification.  The topic is interesting. However, some comments need to be addressed before acceptance. Also, the literature review section misses some important papers.

Abstract

I don’t understand what you mean by a lightweight student model. What is student?

Introduction

There are many VisT and CNNs explain why did you use Swin Transformer and ResNet152V2?

Literature Review

Some important papers are missing related to diabetic retinopathy diagnosis using deep learning methods. Could you please add the following articles?

These articles use DR diagnosis

Detection and classification of red lesions from retinal images for diabetic retinopathy detection using deep learning models

A Lightweight Robust Deep Learning Model Gained High Accuracy in Classifying a Wide Range of Diabetic Retinopathy Images

GabROP: Gabor Wavelets-Based CAD for Retinopathy of Prematurity Diagnosis via Convolutional Neural Networks

DIAROP: Automated Deep Learning-Based Diagnostic Tool for Retinopathy of Prematurity

An active deep learning method for diabetic retinopathy detection in segmented fundus images using artificial bee colony algorithm

A novel approach for diabetic retinopathy screening using asymmetric deep learning features

Could you summarize the literature in a Table.

Methodology:

Please correct the typo “3.1.1. Image Demonising”

What do you mean by Teacher and student Models?

Did you use augmentation techniques?

How did you deal with the class imbalance problem?

Please add the number of images in each class. Please add more details regarding the dataset.

What is the learning rate used? How did you choose the values of the hyperparameters?

Experimental Results

Please validate the performance of the proposed framework on one more dataset.

Author Response

(The authors gave the same response as above.)

Round 2

Reviewer 2 Report

Thanks for addressing my comments

There are some typos all over the manuscript